# Cohort profile: InfCareHIV, a prospective registry-based cohort study of people with diagnosed HIV in Sweden

Christina Carlander ©,[1,2] Johanna Brännström,[2,3] Fredrik Månsson,[4] Olof Elvstam,[5,6] Pernilla Albinsson,[1] Simon Blom,[7] Lena Mattsson,[1] Sanne Hovmöller,[8] Hans Norrgren,[9] Åsa Mellgren,[10] Veronica Svedhem,[1,2] Magnus Gisslén,[11,12] Anders Sönnerborg[1,2,13]

For numbered affiliations see end of article.

**Correspondence to**
Dr Christina Carlander;
christina.carlander@ki.se

## ABSTRACT

**Purpose** The Swedish InfCareHIV cohort was established in 2003 to ensure equal and effective care of people living with HIV (PLHIV) and enable long-term follow-up. InfCareHIV functions equally as a decision support system as a quality registry, ensuring up-to-date data reported in real time.

**Participants** InfCareHIV includes data on >99% of all people with diagnosed HIV in Sweden and up to now 13 029 have been included in the cohort. InfCareHIV includes data on HIV-related biomarkers and antiretroviral therapies (ART) and also on demographics, patient-reported outcome measures and patient-reported experience measures.

**Findings to date** Sweden was in 2015 the first country to reach the UNAIDS (United Nations Programme on HIV/AIDS)/WHO's 90-90-90 goals. Late diagnosis of HIV infection was identified as a key problem in the Swedish HIV-epidemic, and low-level HIV viraemia while on ART associated with all-cause mortality. Increased HIV RNA load in the cerebrospinal fluid (CSF) despite suppression of the plasma viral load was found in 5% of PLHIV, a phenomenon referred to as 'CSF viral escape'. Dolutegravir-based treatment in PLHIV with pre-existing nucleoside reverse transcriptase inhibitor-mutations was non-inferior to protease inhibitor-based regimens. An increase of transmitted drug resistance was observed in the InfCareHIV cohort. Lower efficacy for protease inhibitors was not due to lower adherence to treatment. Incidence of type 2 diabetes and insulin resistance was high in the ageing HIV population. Despite ART, the risk of infection-related cancer as well as lung cancer was increased in PLHIV compared with HIV-negative. PLHIV were less likely successfully treated for cervical precancer and more likely to have human papillomavirus types not included in current HPV vaccines. Self-reported sexual satisfaction in PLHIV is improving and is higher in women than men.

**Future plans** InfCareHIV provides a unique base to study and further improve long-term treatment outcomes, comorbidity management and health-related quality of life in people with HIV in Sweden.

## STRENGTHS AND LIMITATIONS OF THIS STUDY

⇒ Complete nation-wide inclusion of all diagnosed people with HIV.
⇒ High accuracy, completeness, consistency and validity of data.
⇒ Ability to link to other nation-wide healthcare registers.
⇒ Includes patient-reported outcomes.
⇒ Lacking information on socioeconomic data.

## INTRODUCTION

With access to antiretroviral therapy (ART) the overall life expectancy of people living with HIV (PLHIV) is close to that of HIV-seronegative people, as shown in a study including European and North American HIV cohorts.[1] However, studies from the Swedish InfCareHIV cohort show that after a 15-year follow-up period, successfully treated PLHIV in Sweden were three times more likely to die when compared with HIV-seronegative controls.[2] The InfCareHIV cohort is heterogenous, including a large proportion of migrants from 108 countries on all continents, diverse modes of HIV transmission and socioeconomic backgrounds and is gender balanced. The *pol* sequence of the virus is characterised at diagnosis regarding drug resistance mutations and subtype. All known HIV-1 subtypes, many recombinants as well as unique recombinant forms and HIV-2 are represented. Pretreatment HIV drug resistance is a critical aspect that requires global collaborative studies and to minimise its effect, retention in care and optimal adherence to treatment is essential.[3]

The Swedish HIV cohort is ageing and faced with increasing risk of age-related comorbidities. Also, chronic HIV-related inflammation, legacy of pretreatment immunodeficiency and injury and past and present ART may add to the development of comorbidity.[4 5] Additionally, half of PLHIV in Sweden are still late presenters, defined as CD4 count <350

and/or AIDS at time of diagnosis, which has well-known consequences for prognosis and transmission.[6 7] Despite undetectable plasma viral levels, health-related quality of life (HRQoL) is shown to be poorer compared with HIV-seronegative people, in a study from the UK.[8 9] This article describes the prospective Swedish HIV cohort, InfCareHIV, including all diagnosed PLHIV in Sweden, with data dating back to the earliest cases in the 1980s.

## COHORT DESCRIPTION
### Study population
The InfCareHIV cohort was established in 2003 in the two largest cities in Sweden, Stockholm and Gothenburg. These two cities care for about half of all PLHIV in Sweden. The other Swedish clinics joined thereafter and since 2008 the cohort has had complete national coverage, including all 29 clinical HIV centres in Sweden. To date (August 2022), 13 029 PLHIV have been included of whom 8436 are currently in active care (table 1, figure 1). In total 42 PLHIV have been diagnosed with HIV-2 of whom 28 are in active care. In the beginning of the HIV epidemic the majority of PLHIV were born in Sweden, most of whom were men who have sex with men (57.5%) or transmitted through intravenous drug use (26.9%) (online supplemental table 1). The number of migrants with HIV, in particular from sub-Saharan Africa (SSA) and Asia, increased gradually, as did the proportion of women and heterosexually transmitted individuals (online supplemental table 1). A study performed between 2009 and 2012 found 58% of newly diagnosed in Sweden to be late presenters (CD4 <350 cells/µL or AIDS) and 38% to have an advanced infection (<200 cells/µL or AIDS). Ten years later the proportion of late diagnosis remains high (63% and 42% with CD4 <350 and <200, respectively).[6 7]

Of those in active care (August 2022) 39% are women and 1.2% are children below 18 years of age. A majority was born outside Sweden (67%), most commonly in SSA (36%). About half (51%) have stated heterosexual mode of transmission, about one-third are men who have sex with men or bisexual men (31%). Only 4% have stated intravenous drug use as mode of transmission while 3% were mother-to-child transmissions, 1.4% through blood products and for 9.6% transmission mode is unknown or information is missing (table 1). Almost all are on ART (98%), and of them almost everybody (95%) reaches the treatment goal of HIV-RNA <50 copies/mL (98% HIV-RNA <200 copies/mL) in snapshot analysis after at least 6 months of treatment.[10] Of the 4483 people no longer on follow-up in InfCareHIV, 56% are deceased, 34% have emigrated and 3% are lost-to follow-up. InCareHIV also includes undocumented migrants who often end-up among the 3% that are lost-to follow-up (table 1). Retrospective data from before the cohort was established as nation-wide in 2008 has been backlogged, including PLHIV deceased before 2008.

### Coverage and validation of the study population
The national coverage is nearly 100% as all 29 clinical centres that attend PLHIV are included in InfCareHIV, and more than 99% of all diagnosed PLHIV who are living in Sweden are included. This number has been repeatedly validated on an aggregated level, most recently in 2019, against the number of HIV-diagnoses reported to the Public Health Agency of Sweden, a report that is mandatory by Swedish law. The validations found that data regarding HIV diagnoses were even more accurate in InfCareHIV than in the Public Health Agencies records. When transferring between clinics the person keeps their individual InfCareHIV identity number with no data lost.

### Cohort variables
Variables collected and manually registered by a health professional at enrolment include sex at birth, gender identity (added 2021), country of birth, mode of HIV transmission, date of any last negative HIV-test and first positive HIV-test (in Sweden and if relevant abroad), any AIDS diagnoses (eg, pneumocystis pneumonia, oesophageal candidiasis or tuberculosis), confirmed primary HIV infection and suspected country of HIV transmission (online supplemental table 2). Data are collected/updated at each follow-up visit; manually or automatically depending on the type of data and the HIV centre. Data include ART start and stop dates, ART regimen (including doses and mode of administration) and the reason for any change of drug regimens, prophylaxis of opportunistic infections and selected co-medications. Serological and virological data on co-infection with hepatitis C and B virus, weight, date and type of AIDS-defining events, date and cause of death are also included. HIV-RNA (plasma and cerebrospinal fluid (CSF)) and HIV drug resistance (including viral sequences) results, described as both mutations and predicted phenotypic sensitivity, as well as CD4+ and CD8+ T cell counts and CD4/CD8 ratios are either automatically or manually included depending on the HIV centre. HIV-1 subtype and HLA B-5701-allele are registered (online supplemental table 2). Pregnancies and following deliveries are also registered. Socioeconomic data such as civil status, education level or income is not included in InfCareHIV.

### Validation of cohort variables
Validation of data is performed regularly on national level using a Data Quality Index, with the possibility to validate also on clinical and individual level. Data Quality Index for InfCareHIV is currently at 4.9 on a 0–5 scale. Data are also validated through the Distributed Data Management tool which enables data extraction for data sharing and automatically conducts quality assurance checks that signal any incorrect values that may then be manually corrected. This tool is derived and modified from a collaboration with EuroCoord, the European Network of HIV/AIDS Cohort studies to coordinate at European and International Level Clinical Research on HIV/AIDS.[11]

**Table 1**  Description of the InfCareHIV cohort, August 2022

| | In active care | Not active | All |
|---|---|---|---|
| All | 8436 | 4593 | 13 029 |
| Sex | | | |
| Female | 3322 (39.4) | 1042 (22.7) | 4364 (33.5) |
| Birth region | | | |
| Sweden | 2820 (33.4) | 1708 (37.2) | 4528 (34.7) |
| Western Europe except Sweden | 392 (4.6) | 450 (9.8) | 842 (6.5) |
| Eastern Europe and Central Asia | 532 (6.3) | 301 (6.5) | 833 (6.4) |
| Asia and Pacific | 853 (10.1) | 191 (4.2) | 1044 (8.0) |
| Middle East and North Africa | 238 (2.8) | 74 (1.6) | 312 (2.4) |
| Sub-Saharan Africa | 3069 (36.4) | 1006 (21.9) | 4075 (31.3) |
| Latin America and the Caribbean | 370 (4.4) | 179 (3.9) | 549 (4.2) |
| North America | 39 (0.5) | 57 (1.2) | 96 (0.7) |
| Missing | 123 (1.5) | 627 (13.6) | 750 (5.8) |
| Mode of HIV transmission | | | |
| Heterosexual | 4270 (50.6) | 1544 (33.6) | 5814 (44.6) |
| Men who have sex with men/bisexual | 2657 (31.5) | 1707 (37.2) | 4364 (33.5) |
| Intravenous drug use | 353 (4.2) | 740 (16.1) | 1093 (8.4) |
| Mother to child | 254 (3.0) | 23 (0.5) | 277 (2.1) |
| Blood products | 116 (1.4) | 118 (2.6) | 234 (1.8) |
| Unknown/other | 608 (7.2) | 238 (5.2) | 846 (6.5) |
| Missing | 178 (2.1) | 223 (4.9) | 401 (3.1) |
| Year of HIV diagnosis | | | |
| 1979–1996 | 1384 (16.4) | 2482 (54.0) | 3866 (29.7) |
| 1997–2006 | 2209 (26.2) | 867 (18.8) | 3076 (23.6) |
| 2007–2016 | 4367 (41.1) | 971 (21.1) | 4438 (34.1) |
| 2017–2022 | 1196 (14.2) | 124 (2.7) | 1320 (10.1) |
| Missing | 180 (2.1) | 149 (3.2) | 329 (2.5) |
| Level of immunosuppression | | | |
| Nadir CD4 cell count, median (IQR) | 230 (122–370) | 165 (40–330) | 218 (90–359) |
| Missing nadir CD4 cell count, n (%) | 26 (0.3) | 449 (9.8) | 475 (3.6) |
| Last registered CD4 cell count, median (IQR) | 620 (460–813) | 330 (93–560) | 549 (340–750) |
| Antiretroviral therapy | | | |
| On ART | 8305 (98.4) | – | – |
| Treated ≥6 months with HIV-RNA <50 copies/mL | 7719 (94.6) | – | – |
| Treated ≥6 months with HIV-RNA <200 copies/mL | 8267 (98.0) | – | – |
| Seven most common treatment combinations of those on ART (n=8305) | | | |
| Abacavir+lamivudine+dolutegravir | 1650 (19.9) | – | – |
| Emtricitabine+tenofovir(TDF)+dolutegravir | 1256 (15.1) | – | – |
| Lamivudine+dolutegravir | 802 (9.7) | – | – |
| Emtricitabine+tenofovir(TAF)+bictegravir | 719 (8.7) | – | – |
| Emtricitabine+tenofovir(TDF)+efavirenz | 590 (7.1) | – | – |
| Emtricitabine+tenofovir(TAF)+dolutegravir | 574 (6.9) | – | – |
| Emtricitabine+tenofovir(TAF)+rilpivirine | 449 (5.4) | – | – |
| Reason for not being in active care | | | |
| Dead | – | 2575 (56.1) | – |
| Emigrated | – | 1584 (34.5) | – |

Continued

**Table 1** Continued

|  | In active care | Not active | All |
|---|---|---|---|
| Lost to follow-up | – | 150 (3.3) | – |
| Reason not stated | – | 284 (6.2) | – |

Data are numbers and percentages (%) unless otherwise stated. Laboratory data (CD4 cell count and HIV-RNA viral loads) and data regarding antiretroviral therapy refer to latest available data unless otherwise stated. Individual data is generally updated at least every 6 months. People living with HIV diagnosed <1983 were diagnosed retrospectively on biobanked blood once HIV-testing was introduced. Percentages do not always add up to 100 due to rounding. Birth regions according to UNAIDS (United Nations Programme on HIV/AIDS) definitions.
ART, antiretroviral therapy; TAF, tenofovir alafenamide; TDF, tenofovir disoproxil fumarate.

## Ethics

At time of entering HIV care, PLHIV are informed about the registry, after which they can either give verbal consent or opt out. Participants always have the right to exit InfCareHIV, although this has been very uncommon. Participants can request an extract on their data from the registry, free of charge, in accordance with the European General Data Protection Regulation (GDPR 2016/679) and the Swedish Data Protection Act (2018:218).

## Patient and public involvement

A representative from a Swedish organisation for PLHIV (www.hiv-sverige.se) is co-opted to the InfCareHIV steering committee. A web page (www.infcarehiv.se) contains information for both PLHIV, professionals involved in HIV care, researchers and the public. Statistics and treatment results of the individual clinical centres are published openly in an annual report and on the web page.

## Decision support system

The mainstay of InfCareHIV is the clinical decision support system (figure 1). Data on HIV-RNA levels (plasma and CSF), CD4 cell counts (absolute and percentage), drug resistance and previous and current ART are presented through a graphical system. In addition, the legend below the graph presents data regarding

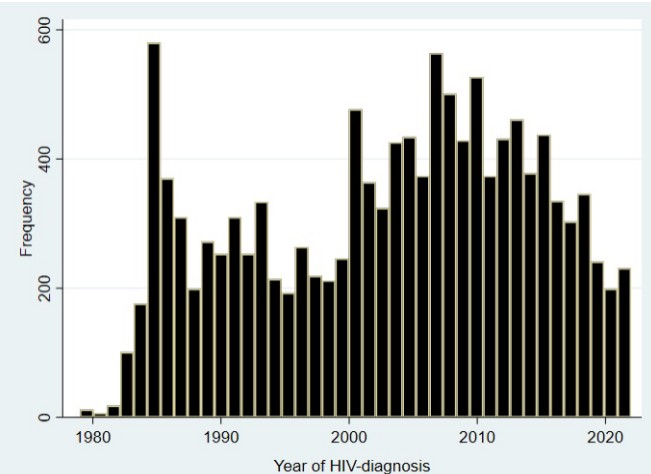

**Figure 1** Year of HIV diagnosis. People with diagnosis <1983 were diagnosed retrospectively on biobanked blood once HIV-testing was introduced.

date of HIV diagnosis, date of first ART and names of the HIV-team consisting of doctor, nurse and counsellor. Periods of pregnancy are also shown in the graph. All other information is easily accessible in the system. Also, since all viral (*pol*) sequences are stored in the database, reanalysis can easily be performed when new drugs or new information about resistance appear. Thus, data generated in clinical care are automatically 'reused' for research and development. The graph gives the clinician a relevant visual summary that is easier to grasp than the medical records, especially in treatment experienced PLHIV with a long history of different ART. Meanwhile the graph is also a pedagogical instrument during patient consultation, facilitating patient-centred care, making it easy to illustrate, for example, the consequences of good or poor treatment adherence. HRQoL is also illustrated in the graph (figure 2). Finally, the graph is used when transferring PLHIV between clinic centres, for expert-consultancy and at multidisciplinary team conferences.

## Quality registry

An important feature of InfCareHIV is to function as a National Quality Registry. The aim of a quality registry is to develop and ensure quality of care systematically and continuously. This allows for comparisons on HIV care at a national, regional and clinical centre level and has led to increased equality of care on national level with diminishing differences in treatment results in Sweden's HIV centres. We believe it is most likely that this contributed to that Sweden in 2015 was the first country to reach the United Nations Programme on HIV/AIDS (UNAIDS)/ WHO's 90-90-90 goal with 90% diagnosed, 90% of them on ART and 90% of them with viral suppression.[10] In concordance with WHO/UNAIDS updated goal, we are now measuring the 95-95-95 goal and have also added 95 goals for HRQoL.

## Health-related quality of life

Systematic quantification of patient-reported outcomes (PROs) will assist the improvement of medical care and HRQoL of PLHIV. In 2011, a 9-item self-reported Health Questionnaire (HQ) was integrated to the registry to be answered annually by PLHIV either via a website or using a computerised or paper version at the outpatient clinic (online supplemental

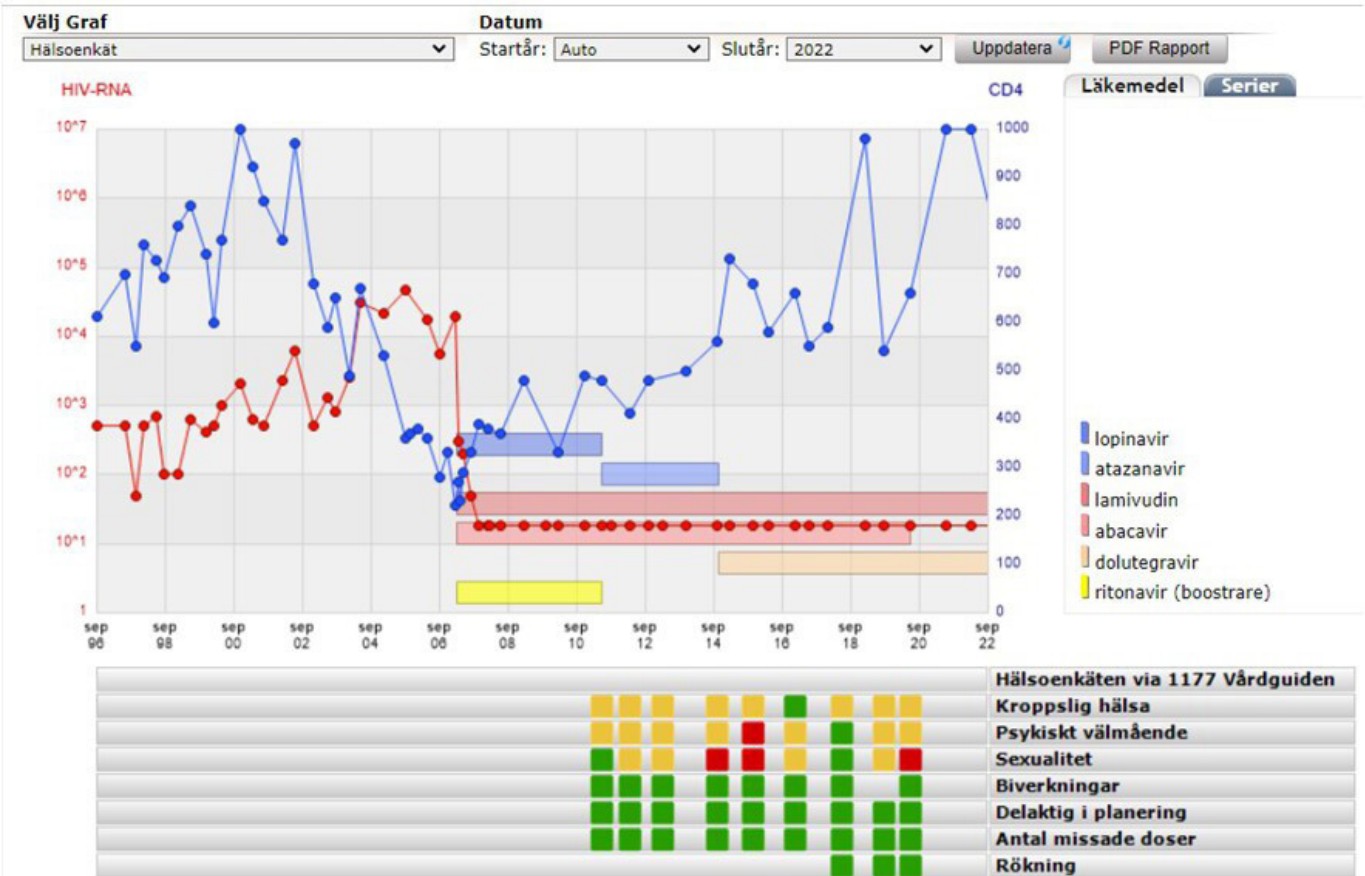

**Figure 2** Clinical decision support system with results from self-reported health questionnaire below the graph. Red symbolises low ratings that need attention (very unsatisfied/unsatisfied), yellow medium ratings (rather unsatisfied/rather satisfied) and green contentment (satisfied/very satisfied), online supplemental table 3. Antal missade doser, missed doses of antiretroviral therapy; Biverkningar, side effects; Delaktig i planering, patient participation in planning of care; Kroppslig hälsa, physical health; Psykiskt välmående, psychological health; Rökning, smoking; Sexualitet, sexual health.

table 3).[12] The questionnaire was developed together with representatives from PLHIV patient organisations. The HQ assesses patient-reported outcome measures (PROMs) regarding physical, psychological and sexual health, self-reported ART adherence, experience of side effects and patient-reported experience measures regarding involvement and satisfaction with care. A question on smoking habits was added in 2017. The questionnaire is electronically available in Swedish and English and in paper version in the eight most common languages in the InfCareHIV cohort. Illiterate patients are offered help by a nurse/interpreter. The results are presented in the Decision support system (figure 2) and used at the patient's routine clinical follow-up visit. The HQ enhances patient-centred HIV care by focusing the consultation on the patients current needs. The red signal symbolises low ratings (*very unsatisfied/unsatisfied),* that need attention, yellow medium ratings (*rather unsatisfied/rather satisfied*) and green contentment (*satisfied/very satisfied),* (online supplemental table 3). There is a standardised clinical manual for the health personnel with suggestions on how to manage yellow and red

alerts (eg, screening for depression and asking about intimate partner violence if psychological well-being gets a yellow or red alert). The HQ is validated by content validity and test–retest reliability and has also been evaluated as a tool for longitudinal follow-up of trends in PROs.[12–14] Our fourth 95 goal is that 95% of all PLHIV should perform the health questionnaire and that 95% of PLHIV should have an ART regimen without experience of side effects. The results from the HQ are also used in research, more on that below.

HIV-related stigma and discrimination are major obstacles for reaching good HRQoL. In close relation to InfCareHIV we developed a 12-item HIV stigma scale, a short version of the commonly used 40-item HIV Stigma Scale by Berger *et al.*[15] It is a valid and reliable instrument for the measurement of enacted, anticipated and internalised stigma and crucial for mapping trends in the prevalence of HIV-related stigma and tracking the effectiveness of stigma-reducing interventions.[15 16]

**Biobank**
In the major clinical centres blood plasma/serum/liquor samples are collected, and in selected cases also

peripheral blood mononuclear cells, HIV isolates, CSF and CSF cells in biobanks. These biobanks are separate from the cohort but linking between the cohort data and the biobanks can be performed (after ethical permission) for translational studies.

## Selected findings from the past 10 years to date

### Studies on late presentation

In a national prospective study, using InfCareHIV linked to patient study forms, Brännström *et al* found late diagnosis of HIV infection to be a key problem in the Swedish HIV-1 epidemic, where more than half of the patients were diagnosed late, but the majority could have been diagnosed earlier with a more efficient healthcare system.[6 17 18] Most of the patients experienced barriers to HIV testing, but less so if the HIV-test was offered through screening programmes or by a healthcare professional rather than having to be self-initiated.[19 20] Results from the study have been adopted by the Public Health Agency of Sweden and have had implications for the Swedish governmental National strategy against HIV/AIDS and other infectious diseases.[21]

### Studies on comorbidity and mortality

Elvstam *et al* reported an association between low-level HIV viraemia while on ART and all-cause mortality,[22] and has further explored the effects of detectable viraemia during ART by linking InfCareHIV to National Health registries and analysing stored biobank samples.[23–27] Malmström *et al* studied risk of cancer by HIV status in Sweden for three decades, finding that PLHIV have a remaining increased risk of infection-related cancer despite ART, while lung cancer was the only non-infection-related cancer increased in PLHIV.[28] By linking InfCareHIV to the National Cervical Cancer Screening registry, Carlander *et al* showed that PLHIV are less likely to have successful treatment of cervical precancer and more likely to have human papillomavirus (HPV) types not included in current HPV vaccines.[29–32] These results were used when the national cervical-cancer-screening recommendation was updated in 2022.[33] Bratt *et al* showed that the incidence of type 2 diabetes and insulin resistance is high in the ageing HIV population, where comorbidities are common and associated whereas there was no association found for HIV-related factors.[34] Möller *et al* showed that well-treated PLHIV are not at higher odds of severe COVID-19 compared with HIV-negative people after controlling for age and comorbidity.[35]

### Studies on HIV infection of the central nervous system

Gisslén *et al* have used InfCareHIV in a substantial number of studies exploring CSF viral load and other biomarkers in different settings of untreated and treated HIV.[36–54] Viral load is normally one log lower in CSF than in plasma in untreated HIV,[55] but CSF exceeds plasma HIV RNA in approximately 15% of patients, with considerable variations between different disease stages.[36] Central nervous system (CNS) infection is generally well controlled by systemic suppressive ART,[56] although in approximately 5% the HIV RNA load was increased in the CSF despite suppression of the plasma viral load, a phenomenon referred to as 'CSF viral escape'.[41] This phenomenon is most often transient, not associated with any symptoms and comparable to plasma viral blips[57] which occur in about the same frequency.[58] Several studies have indicated that a stable, permanent infection of cells in the CNS is established later than in systemic viral reservoirs,[40] which has implications when exploring HIV eradication strategies.[59]

### Studies on elite controllers

Sönnerborg *et al* have extensively studied, since the 1990s, long-term non-progressors and elite controllers (EC), selected from the InfCareHIV cohort. Specific immunological[60] and metabolic features have been described.[61] Also, a naturally occurring dipeptide was found to be enhanced among EC and to possess antiretroviral properties, acting as both an entry inhibitor and an RT-inhibitor.[62]

### Studies on drug resistance mutations

Sönnerborg *et al* have also studied HIV drug resistance, both transmitted and acquired, and the molecular HIV epidemiology. Among several key findings, an increase of transmitted drug resistance in the InfCareHIV cohort[3] was reported. A pronounced HIV-1 subtype heterogeneity, including all known subtypes, many circulating recombinant forms and unique recombinant forms was also described.[63] Sörstedt *et al* studied the effect of dolutegravir-based treatment in PLHIV with pre-existing NRTI-mutations and found a non-inferior effect compared with protease inhibitor-based regimens[64] and has also explored viral blips during ART.[58]

### Studies on patient-reported outcomes and experiences

Svedhem *et al* validated the Health Questionnaire and demonstrated that self-reported adherence in the HQ was correlated to viral suppression and described determinants of optimal ART-adherence in the cohort.[12] Svedhem *et al* have shown that the assessment of PROMs is an important tool to ensure the long-term adherence to treatment, improvement in quality of life and evaluate side effects on HIV treatment.[12 14] PROMs have also successfully been used in virological molecular modelling showing that lower affinity for protease inhibitors to HIV-1C protease do not depend on lower adherence to treatment among people infected with subtype C.[13] Tyrberg *et al* compared plasma drug levels of ART, potential drug–drug interactions and side-effects in PLHIV aged ≥65 years of age, with controls ≤49 years of age, and found differences in drug concentrations and reported side effects between groups.[65]

Mellgren *et al* demonstrated that the experience of side effects of ART declined significantly in the cohort during 2011–2017 and that experiences of side effects were diverse and associated with both self-reported physical and psychological health.[14] Two studies on self-reported

sexuality by Mellgren *et al* found that self-reported sexual satisfaction in PLHIV improved annually and that women were more satisfied with their sexual life compared with men.[66 67] In women living with HIV, satisfaction with sexual life was associated with self-reported psychological health and experiences of side effects. Carlander *et al* showed that over the past 20 years access to employment has increased in PLHIV although remaining lower compared with HIV-negative, even after controlling for migrant and socioeconomic status.[68]

## International collaborations

InfCareHIV has collaborated or collaborates with several international HIV cohorts such as EuroCoord, EuroSIDA, RESPOND, PENTA, COHERE, CASCADE, CHAIN, NEAT, EuResist, EuCare, UCSF CSF Cohort and CARE.

## Future plans

InfCareHIV provides a unique base to continue study long-term treatment, comorbidities and HRQoL of people with HIV in Sweden. Half of all PLHIV is still diagnosed late and studies that can help improve HIV-testing guidelines and HIV awareness are essential. Attempts are made to develop more precise bioinformatics tools for assessment of time of infection and the undiagnosed population. Pretreatment HIV drug resistance is a critical aspect that requires global collaboration and long-term follow-up. Despite most PLHIV reaching the treatment goal of undetectable viral level, patient reported health is still poorer than for HIV-negative people and studies on how to improve HRQoL in PLHIV and minimise stigma should be prioritised. With our ageing cohort, studies on comorbidity become more important and specially to assess what comorbidities are associated with normal ageing and what comorbidities may be associated with ART or HIV-induced chronic inflammation. Multiomics using samples and clinical information is presently evaluated to identify biomarkers for prediction of comorbidities, to characterise the evolution of CNS inflammation and injury[45] and for future selection of patients in HIV cure attempts.

## Strengths and limitations

The main strength of the InfCareHIV cohort is that it includes all diagnosed PLHIV in Sweden. The accuracy, completeness, consistency and validity is very high and consequently InfCareHIV has the highest certification level among Swedish Health Quality registries according to the Swedish Association of Local Authorities and Regions.[69] The personal identity number given to all Swedish residents enables linkage of data to any Swedish population or health registry after appropriate application, providing opportunities to answer a wide range of research questions. Also, the automatic transfer of key laboratory data, including viral sequences, and the possibility of linking cohort data and biobanks allow easy reuse of information obtained in clinical care for research purposes. Some variables in InfCareHIV (eg, comorbidity,

weight and self-reported smoking status that has been added in later years) lacks in coverage which currently limits their use in research but linking to other health registries for the collection of this data is then a possibility. The aim is for all PLHIV to be invited to answer the self-reported health questionnaire annually to facilitate patient centred care and improve HRQOL, although currently we reach only about 34%, and it is a prioritised matter to improve this number.

**Author affiliations**
[1]Department of Infectious Diseases, Karolinska University Hospital, Stockholm, Sweden
[2]Department of Medicine Huddinge, Karolinska Institute, Stockholm, Sweden
[3]Department of Infectious Diseases/Venhälsan, Södersjukhuset, Stockholm, Sweden
[4]Department of Clinical Sciences, Lund University, Infectious Diseases Research Unit, Malmo, Sweden
[5]Department of Translational Medicine, Lund University, Lund, Sweden
[6]Department of Infectious Diseases, Växjö Central Hospital, Växjö, Sweden
[7]HIV Sweden, Stockholm, Sweden
[8]Department of Infectious Diseases, Sunderby Hospital, Lulea, Sweden
[9]Department of Clinical Sciences, Lund University Faculty of Science, Lund, Sweden
[10]Department of Infectious Diseases, Sahlgrenska University Hospital, Gothenburg, Sweden
[11]Department of Infectious Diseases, Institute of Biomedicine, University of Gothenburg Sahlgrenska Academy, Gothenburg, Sweden
[12]Department of Infectious Diseases, Region Västra Götaland, Sahlgrenska University Hospital, Gothenbrug, Sweden
[13]Department of Laboratory Medicine, Karolinska Institute, Stockholm, Sweden

**Acknowledgements** First of all, the authors would like to thank all people living with HIV in Sweden enrolled in the cohort. We also thank all clinical centres that work daily with the inclusion and update of data in InfCareHIV.

**Contributors** JB, FM, OE, PA, SB, LM, SH, HN, AM, VS, MG and AS planned and designed the cohort profile description. CC drafted the manuscript. JB, FM, OE, PA, SB, LM, SH, HN, AM, VS, MG and AS critically revised the manuscript and approved the final version. CC, as guarantor, accepts full responsibility of the work, had access to the data, and controlled the decision to publish.

**Funding** The authors have not declared a specific grant for this research from any funding agency in the public, commercial or not-for-profit sectors.

**Competing interests** CC has received lecture, moderator and advisory board fees from GSK/ViiV, Gilead Sciences and MSD and an unrestricted Nordic Fellowship Grant from Gilead Sciences Nordic. JB has received lecture and advisory board fees from GSK/ViiV and Gilead. FM has received lecture and advisory board fees from GSK/ViiV, AstraZeneca and Gilead Sciences. OE has received a grant to his institution from Pfizer and honoraria as speaker from Gilead Sciences. HN has received advisory board fees from Gilead and AbbVie. AM has received lecture and advisory board fees from GSK/ViiV, Gilead Sciences and Pfizer. MG has received research grants from Gilead Sciences and Janssen-Cilag and honoraria as speaker, DSMB committee member and/or scientific advisor from Amgen, AstraZeneca Biogen, Bristol-Myers Squibb, Gilead Sciences, GlaxoSmithKline/ViiV, Janssen-Cilag, MSD, Novocure, Novo Nordic, Pfizer and Sanofi. AS has received research grants from Gilead Sciences and honoraria as speaker, DSMB committee member and/or scientific advisor from AstraZeneca, Bristol-Myers Squibb, Gilead Sciences, GlaxoSmithKline/ViiV and MSD. All other authors declare no competing interests.

**Patient and public involvement** Patients and/or the public were involved in the design, or conduct, or reporting, or dissemination plans of this research. Refer to the Cohort description section for further details.

**Patient consent for publication** Not applicable.

**Ethics approval** This cohort study has been approved by the Regional Ethical committees in Sweden (Dnr 532-11, Dnr 2018/11-31/2, Dnr 2022-02897-02).

**Provenance and peer review** Not commissioned; externally peer reviewed.

**Data availability statement** Data are available upon reasonable request. Data can be made available upon reasonable request and after ethical approval.

**ORCID iD**
Christina Carlander http://orcid.org/0000-0001-9962-5964

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
