## [Reviewer comments · BMJ Open]

ARTICLE DETAILS

TITLE (PROVISIONAL)	Cohort profile: InfCareHIV, a prospective registry-based cohort study of people with diagnosed HIV in Sweden
AUTHORS	Carlander, Christina; Brännström, Johanna; Mansson, Fredrik; Elvstam, Olof; Albinsson, Pernilla; Blom, Simon; Mattsson, Lena; Hovmöller, Sanne; Norrgren, Hans; Mellgren, Aasa; Svedhem, Veronica; Gisslén, Magnus; Sonnerborg, Anders

VERSION 1 – REVIEW

REVIEWER	Lampe, F University College London, Research Department of Infection and Population Health
REVIEW RETURNED	15-Dec-2022

GENERAL COMMENTS	This paper describes a national cohort of people with HIV in Sweden, covering all HIV clinics nationally, with an opt-out consent procedure. A comprehensive description of the data and procedures are given, I have just a few points for clarification. It would be useful to add to the introduction a brief overview of the HIV epidemic in Sweden, in terms of numbers affected, trends over time in numbers/diagnoses, late diagnosis etc. Also in the introduction, please make it clear when quoting information from the literature what region/country/setting it refers to. A paragraph on ethics is given, but there is no reference to an ethics committee approval. Please give details on this or why it was not necessary. In Table 1, please give information (in a footnote or in the text) to clarify which measures have been used – for example is it the latest value that is used for ART and VL, and what is the window period? If individuals transfer between clinics in Sweden, I am presuming they are still tracked as the same person in the database, using the personal identity number? The addition of self-reported information to the database is interesting and important. Does this include validated tools, and if so what are they? It would also be useful to have additional information on how this is used clinically. How are the different alert levels (red, green etc) defined and is there a standardised procedure for what is done clinically on this basis. Also, to what extent are these self-reported factors used as a clinical tool, or are they more a research tool? The paper includes a large section of text listing previous results
--

	from the cohort. The text is quite dense, and it may benefit from being put under subheadings, or being more selective. Perhaps using a graph or table or two from previous papers would help to highlight key results. Also, it would be useful to highlight more the implications of some of the findings and emphasise their importance for health/research/policy or development of the cohort. There is little information on trends over time within the cohort for key outcomes. This paper describes a national cohort of people with HIV in Sweden, covering all HIV clinics nationally, with an opt-out consent procedure. A comprehensive description of the data and procedures are given, I have just a few points for clarification. It would be useful to add to the introduction a brief overview of the HIV epidemic in Sweden, in terms of numbers affected, trends over time in numbers/diagnoses, late diagnosis etc. Also in the introduction, please make it clear when quoting information from the literature what region/country/setting it refers to. A paragraph on ethics is given, but there is no reference to an ethics committee approval. Please give details on this or why it was not necessary. In Table 1, please give information (in a footnote or in the text) to clarify which measures have been used – for example is it the latest value that is used for ART and VL, and what is the window period? If individuals transfer between clinics in Sweden, I am presuming they are still tracked as the same person in the database, using the personal identity number? The addition of self-reported information to the database is interesting and important. Does this include validated tools, and if so what are they? It would also be useful to have additional information on how this is used clinically. How are the different alert levels (red, green etc) defined and is there a standardised procedure for what is done clinically on this basis. Also, to what extent are these self-reported factors used as a clinical tool, or are they more a research tool? The paper includes a large section of text listing previous results from the cohort. The text is quite dense, and it may benefit from being put under subheadings, or being more selective. Perhaps using a graph or table or two from previous papers would help to highlight key results. Also, it would be useful to highlight more the implications of some of the findings and emphasise their importance for health/research/policy or development of the cohort. There is little information on trends over time within the cohort for key outcomes.
--	--

REVIEWER	Kroch, Abigail Ontario HIV Treatment Network
REVIEW RETURNED	11-Jan-2023

GENERAL COMMENTS	The manuscript describes the Swedish HIV Cohort, including examples of recent publications and findings. The manuscript is well organized, concise and written clearly. The description of the cohort is full and includes adequate technical
---

	description of the recruitment strategy and process. The description of data collection and inclusion of variables is very helpful. The population description includes all relevant information. The summary of recent findings is quite interesting. The manuscript demonstrates well the strength of a cohort study. The expansion of the study to include clinical decision making and PROMs is impressive.
--	---

REVIEWER	Rodrigues, Rashmi St Johns Medical College, Community Health
REVIEW RETURNED	19-Jan-2023

GENERAL COMMENTS	I commend the authors on their effort to capture HIV-related data in Sweden and for your willingness to share it collaboratively. While the manuscript addresses all essential concerns regarding data collection, availability, privacy and confidentiality, and ethics, I have a few comments for your consideration. Please see below: Abstract: Ok Introduction: Line 88 'Meanwhile half of all PLHIV are still diagnosed late with well-known consequences for prognosis and transmission. The sentence is not very clear especially what late means in the context of HIV management in Sweden. Cohort description: Line 113 'Retrospective data from before the cohort was established (1983-2007) have been backlogged including PLHIV deceased before 2008- is this discrepancy because the other clinics joined the cohort by 2008? Coverage and Validation: Is there any specific method used for InfCareHIV diagnosis data validation against the data reported to the Swedish public health agency? – is this done electronically and is it only the aggregate numbers that are validated or is each record individually validated? Also, this validation is only for data concerning diagnosis? Cohort variables: Is the data fed into the system directly at the 'point of care'? Are education, marital status, occupation and socioeconomic status/ income, and housing also included in the database? Is contact tracing done? Comorbidities- is tuberculosis also included? Or is it not a common comorbidity in PLHIV in Sweden?
--

VERSION 1 – AUTHOR RESPONSE

Reviewer: 1

Dr. F Lampe, University College London

Comments to the Author:

This paper describes a national cohort of people with HIV in Sweden, covering all HIV clinics nationally, with an opt-out consent procedure. A comprehensive description of the data and procedures are given, I have just a few points for clarification.

Author reply: *We thank reviewer 1 for this supportive comment.*

Reviewer 1 comment 1: It would be useful to add to the introduction a brief overview of the HIV epidemic in Sweden, in terms of numbers affected, trends over time in numbers/diagnoses, late diagnosis etc.

Author reply: *Thank you for this relevant comment. A brief overview of the HIV epidemic in Sweden (including numbers affected, trends over time in diagnoses and late diagnosis) has now been added to the manuscript (please see Cohort description page 4, lines 107-117, Figure 1, and Supplementary Table 1)*

Reviewer 1 comment 2: Also in the introduction, please make it clear when quoting information from the literature what region/country/setting it refers to.

Author reply: *Thank you for this valid comment. This has now been clarified in the manuscript (please see Introduction page 3).*

Reviewer 1 comment 3: A paragraph on ethics is given, but there is no reference to an ethics committee approval. Please give details on this or why it was not necessary.

Author reply: *Thank you for this important comment. There are several ethic committee approvals. This has been clarified under Ethics (please see Ethics page 6, lines 183-184).*

Reviewer 1 comment 4: In Table 1, please give information (in a footnote or in the text) to clarify which measures have been used – for example is it the latest value that is used for ART and VL, and what is the window period?

Author reply: *Thank you for this comment. Lab data (CD4 cell count and HIV-RNA viral loads) and data regarding antiretroviral therapy in Table 1 refer to latest available data unless otherwise stated. Individual data is generally updated at least every six months. This has now been updated in a footnote under Table 1 (please see Table 1, page 19, lines 700-702).*

Reviewer 1 comment 5: If individuals transfer between clinics in Sweden, I am presuming they are still tracked as the same person in the database, using the personal identity number?

Author reply: *Thank you for this relevant comment. The transfer is handled through the InfCareHIV platform. When an individual is transferred between clinics, health personal at the former clinic will transfer the individual, through InfCareHIV, to the new clinic with no data lost in InfCareHIV. The individual will keep their personal InfCareHIV identity number when transferring (as well as of course their national personal identity number). This has now been clarified in the manuscript (please see page 5, lines 140-142).*

Reviewer 1 comment 6:

The addition of self-reported information to the database is interesting and important. Does this include validated tools, and if so what are they? It would also be useful to have additional information

on how this is used clinically. How are the different alert levels (red, green etc.) defined and is there a standardised procedure for what is done clinically on this basis. Also, to what extent are these self-reported factors used as a clinical tool, or are they more a research tool?

Author reply: Thank you for this important comment. The self-reported questionnaire is indeed an important tool both in the clinic and as a research tool. The questionnaire has been validated by content validity and test-retest reliability and has also been evaluated as a tool for longitudinal follow up of trends in PROs (Marrone et al. High Concordance between Self-Reported Adherence, Treatment Outcome and Satisfaction with Care Using a Nine-Item Health Questionnaire in InfCareHIV. *PloS one*. 2016;11(6):e0156916. Häggblom et al. Virological failure in patients with HIV-1 subtype C receiving antiretroviral therapy: an analysis of a prospective national cohort in Sweden. *Lancet HIV*. Apr 2016;3(4):e166-74.)

The questionnaire is used clinically as a tool to support health-related quality of life in PLHIV. The aim is for the questionnaire to be offered to PLHIV once annually. The questionnaire is followed up by doctor or nurse at the out-patient department. The alert levels are defined as red (very unsatisfied/unsatisfied) yellow (rather unsatisfied/rather satisfied) or green (satisfied/very satisfied), please see Supplementary table 3. There is a standardized clinical manual on how to handle the answers (for example screening for depression if psychological well-being gets a red alert) . The manual is currently being updated and the new version will be available at for example the InfCareHIV website. This has now been clarified in the manuscript (please see pages 7-8, lines 238-244). The result from the self-reported questionnaire is also used in research (please see pages 10-11, lines 324-346).

Reviewer 1 comment 7:

The paper includes a large section of text listing previous results from the cohort. The text is quite dense, and it may benefit from being put under subheadings, or being more selective. Perhaps using a graph or table or two from previous papers would help to highlight key results. Also, it would be useful to highlight more the implications of some of the findings and emphasise their importance for health/research/policy or development of the cohort.

Author reply: Thank you for pointing this out. The Results section is now updated with subheadings and implications of findings has been added (please see pages 8-11, lines 265-346).

Reviewer 1 comment 8:

There is little information on trends over time within the cohort for key outcomes.

Author reply: Thank you for commenting on the relevance of trends over time. So far there is little published data on trends over time for HIV epidemiology in Sweden. It is the intent of the Steering Committee to enable such publication as soon as possible. Figure 1 has been added which illustrates the number of diagnoses over time (please see Figure 1)

Reviewer: 2

Dr. Abigail Kroch, Ontario HIV Treatment Network

Comments to the Author:

The manuscript describes the Swedish HIV Cohort, including examples of recent publications and findings. The manuscript is well organized, concise and written clearly. The description of the cohort is full and includes adequate technical description of the recruitment strategy and process. The description of data collection and inclusion of variables is very helpful. The population description includes all relevant information. The summary of recent findings is quite interesting. The manuscript demonstrates well the strength of a cohort study. The expansion of the study to include clinical decision making and PROMs is impressive.

Author reply: *We thank reviewer 2 for this highly supportive summary.*

Reviewer: 3

Dr. Rashmi Rodrigues, St Johns Medical College, Karolinska Institute

Comments to the Author:

Dear Authors,

I commend the authors on their effort to capture HIV-related data in Sweden and for your willingness to share it collaboratively.

While the manuscript addresses all essential concerns regarding data collection, availability, privacy and confidentiality, and ethics, I have a few comments for your consideration. Please see below:

Author reply: *We thank reviewer 3 for this very positive summary.*

Reviewer 3 comment 1:

Abstract: Ok

Introduction:

Line 88 'Meanwhile half of all PLHIV are still diagnosed late with well-known consequences for prognosis and transmission.

The sentence is not very clear especially what late means in the context of HIV management in Sweden.

Author reply: *Thank you for this comment. Late refers to late presenters, defined as CD4 count<350 and/or AIDS at time of HIV diagnosis. This has now been clarified in the manuscript (please see page 3, lines 91-92).*

Reviewer 3 comment 2: Cohort description:

Line 113 'Retrospective data from before the cohort was established (1983-2007) have been backlogged including PLHIV deceased before 2008- is this discrepancy because the other clinics joined the cohort by 2008?

Author reply: *Thank you for this comment. Yes, given that the cohort was nation-wide from 2008, some clinics were included as early as 2003, data preceding these dates has been backlogged. InfCareHIV includes data on PLHIV diagnosed from 1983 (a few retrospectively from 1979) and onwards. This has now been clarified in the manuscript (please see page 4, lines 130-131).*

Reviewer 3 comment 3: Coverage and Validation: Is there any specific method used for InfCareHIV diagnosis data validation against the data reported to the Swedish public health agency? – is this done electronically and is it only the aggregate numbers that are validated or is each record individually validated? Also, this validation is only for data concerning diagnosis?

Author reply: *Thank you for this highly valid question. Regarding the validation of the number of people diagnosed and the number of new diagnoses annually, this is regularly validated against the Swedish Public Health Agency, and against Infection Control Stockholm (the county of Stockholm includes about half of all PLHIV in Sweden) on an aggregated level. Regarding the validation of data on cohort variables this is performed regularly on national level using a Data Quality Index, with the possibility to validate also on clinic and individual level. Data Quality Index for InfCareHIV is currently at 4.9 on a 0-5 scale. Data are also validated through the Distributed Data Management tool which enables data extraction for data sharing and automatically conducts quality assurance checks that signal any incorrect values that may then be manually corrected. This tool is derived and modified from a collaboration with EuroCoord, the European Network of HIV/AIDS Cohort studies to coordinate at European and International Level Clinical Research on HIV/AIDS.*

This has now been clarified in the manuscript (please see page 4-5, lines 136-138 and pages 5-6, lines 165-173).

Reviewer 3 comment 4: Cohort variables:

Is the data fed into the system directly at the 'point of care?'

Author reply: Thank you for this important question. Yes, most data is fed into the system directly at point of care. Some lab-data is imported directly from the lab at the largest clinics. This has now been clarified in the manuscript (please see page 5, lines 150-152).

Reviewer 3 comment 5: Are education, marital status, occupation and socioeconomic status/income, and housing also included in the database?

Author reply: Thank you for this very relevant comment. No, socioeconomic status is not included in the database. However, with the use of the personal identification number the database can be linked (after consent from the Swedish Ethical Review Authority) to the databases of Statistics Sweden which includes the LISA database (including data on civil status, education, occupation, income etc.). This has been performed in studies on for example studies on employment (Carlander C, Wagner P, Yilmaz A, Sparén P, Svedhem V. Employment by HIV status, mode of HIV transmission and migrant status: a nation-wide population-based study. *AIDS* (London, England). 2021;35(1):115-23) and in studies on the risk of cancer in PLHIV in Sweden (Malmström S, Wagner P, Yilmaz A, Svedhem V, Carlander C. Failure to restore CD4+ cell count associated with infection-related and noninfection-related cancer. *AIDS*. 2022 Mar 1;36(3):447-457. doi: 10.1097/QAD.0000000000003117. PMID: 34711738). This has now been updated in the manuscript (please see page 5, lines 162-163).

Reviewer 3 comment 6: Is contact tracing done?

Author reply: Yes, contact tracing is performed at the clinics after HIV diagnosis but the result is not recorded in InfCareHIV as it would include sensitive data on people that have not given their consent.

Reviewer 3 comment 7: Comorbidities- is tuberculosis also included? Or is it not a common comorbidity in PLHIV in Sweden?

Author reply: Thank you for this comment. Tuberculosis is included among AIDS-diagnoses. This has now been clarified in the manuscript (please see page 5, lines 148-150).

VERSION 2 – REVIEW

REVIEWER	Lampe, F University College London, Research Department of Infection and Population Health
REVIEW RETURNED	06-Mar-2023
GENERAL COMMENTS	The reviewer comments have been addressed